# Preclinical Characterization of the Distribution, Catabolism, and Elimination of a Polatuzumab Vedotin-Piiq (POLIVY^®^) Antibody–Drug Conjugate in Sprague Dawley Rats

**DOI:** 10.3390/jcm10061323

**Published:** 2021-03-23

**Authors:** Victor Yip, M. Violet Lee, Ola M. Saad, Shuguang Ma, S. Cyrus Khojasteh, Ben-Quan Shen

**Affiliations:** 1Preclinical and Translational Pharmacokinetics and Pharmacodynamics, Genentech Inc., 1 DNA Way, South San Francisco, CA 94080, USA; victory@gene.com; 2BioAnalytical Sciences, Genentech Inc., 1 DNA Way, South San Francisco, CA 94080, USA; lee.manjui@gene.com (M.V.L.); ola@gene.com (O.M.S.); 3Drug Metabolism and Pharmacokinetics, Genentech Inc., 1 DNA Way, South San Francisco, CA 94080, USA; mas11@gene.com (S.M.); khojasteh.cyrus@gene.com (S.C.K.)

**Keywords:** antibody–drug conjugate (ADC), distribution, catabolism, and elimination (DME), polatuzumab vedotin (POLIVY), monomethyl auristatin E (MMAE), mass balance

## Abstract

Polatuzumab vedotin (or POLIVY^®^), an antibody–drug conjugate (ADC) composed of a polatuzumab monoclonal antibody conjugated to monomethyl auristatin E (MMAE) via a cleavable dipeptide linker, has been approved by the United States Food and Drug Administration (FDA) for the treatment of diffuse large B-cell lymphoma (DLBCL). To support the clinical development of polatuzumab vedotin, we characterized the distribution, catabolism/metabolism, and elimination properties of polatuzumab vedotin and its unconjugated MMAE payload in Sprague Dawley rats. Several radiolabeled probes were developed to track the fate of different components of the ADC, with ^125^I and ^111^In used to label the antibody component and ^3^H to label the MMAE payload of the ADC. Following a single intravenous administration of the radiolabeled probes into normal or bile-duct cannulated rats, blood, various tissues, and excreta samples were collected over 7–14 days post-dose and analyzed for radioactivity and to characterize the metabolites/catabolites. The plasma radioactivity of polatuzumab vedotin showed a biphasic elimination profile similar to that of unconjugated polatuzumab but different from unconjugated radiolabeled MMAE, which had a fast clearance. The vast majority of the radiolabeled MMAE in plasma remained associated with antibodies, with a minor fraction as free MMAE and MMAE-containing catabolites. Similar to unconjugated mAb, polatuzumab vedotin showed a nonspecific distribution to multiple highly perfused organs, including the lungs, heart, liver, spleen, and kidneys, where the ADC underwent catabolism to release MMAE and other MMAE-containing catabolites. Both polatuzumab vedotin and unconjugated MMAE were mainly eliminated through the biliary fecal route (>90%) and a small fraction (<10%) was eliminated through renal excretion in the form of catabolites/metabolites, among which, MMAE was identified as the major species, along with several other minor species. These studies provided significant insight into ADC’s absorption, distribution, metabolism, and elimination (ADME) properties, which supports the clinical development of POLIVY.

## 1. Introduction

The development of antibody–drug conjugates (ADCs) has accelerated in recent years, resulting in many advancements to this class of therapeutic molecules [1]. Polatuzumab vedotin, which was approved for treating diffuse large B-cell lymphoma (DLBCL), consists of an immunoglobulin G1 (IgG1) monoclonal antibody (mAb) against the antigen Cluster of Differentiation 79B (CD79b, polatuzumab) conjugated with a payload of monomethyl auristatin E (MMAE, vedotin) using a protease-labile linker, namely, maleimidocaproyl-valine-citrulline-p-aminobenzyloxycarbonyl (MC-vc-PAB) [2,3].

The pharmacokinetics (PK) of polatuzumab vedotin in rodents and cynomolgus monkeys were described by Li et al. [4], who showed that the concentration–time profile of polatuzumab vedotin was very similar to that of unconjugated polatuzumab antibodies, with a short distribution phase followed by a long elimination phase. However, the characterization of the absorption, distribution, metabolism, and elimination (ADME) properties of polatuzumab vedotin has not been reported. There is only limited ADME information for other ADCs available in the literature [5,6]. Due to the fact that ADCs contain potent cytotoxic drug payloads, the ADME characterization plays an important role in ADC development, as data from these studies offer insight into the potential of drug–drug interactions (DDIs), organ impairment, and other safety assessments.

Unlike the therapeutic antibodies, where they are often degraded into amino acids, small peptides, or small carbohydrates that are readily eliminated by renal excretion or return to the nutrient pool with minimal biological effects or safety concerns, ADCs contain a potent cytotoxic agent and are structurally more complex. Therefore, in addition to the characterization of the antibody and the cytotoxic payload, the understanding of linker stability is also critical, as a premature release of the payload can cause systemic toxicity [7,8,9].

For oncology indications, ADCs are likely to be used in combination with other chemotherapy agents that may interact with various cytochrome P450 (CYP) enzymes and drug transporters. Therefore, identifying the key catabolites of the ADC is valuable for assessing potential DDIs, determining the key drivers for efficacy and toxicity, and informing on which key analytes should be measured in a clinical setting.

There are various approaches that are used to characterize the ADME properties of ADCs [10]; some groups have used an imaging approach to track the payload delivery, which is less invasive and can be visualized in real time [11,12]. Others took a different approach to understand the disposition of each ADC component via tissue harvesting, as smaller tissues might be missed using the previous approach [6,13].

In this study, we systemically characterized the distribution, catabolism, and elimination (DME) properties (no absorption with intravenous dosing) of polatuzumab vedotin and MMAE in rats using multiple radiolabeled probes to track the distribution, metabolism/catabolism, and elimination of polatuzumab vedotin, as well as the unconjugated MMAE payload. Rats were chosen as the model species as they do not cross-bind with polatuzumab vedotin, thus offering a general DME profile that is similar to patients who would have been dosed clinically in the linear range. These data provided insights into the DME properties of polatuzumab vedotin to support the clinical assessment of the potential DDIs.

## 2. Materials and Methods

### 2.1. Antibodies and ADCs

The polatuzumab and polatuzumab vedotin used for the in vivo studies presented here were generated at Genentech Inc. (South San Francisco, CA, USA). The unconjugated polatuzumab is a humanized monoclonal IgG1 antibody that binds to human CD79b and does not cross-react with rodent CD79b. Polatuzumab vedotin was made as described previously [14].

The conjugation conditions were chosen to achieve an average drug-to-antibody ratio (DAR) of approximately 3.4–3.5. The ADC protein concentrations were calculated using absorbance at 280 nm (a 320 nm reference) and the molar extinction coefficient of the antibody. The average DARs were calculated from the integrated areas of the DAR species, which were resolved using hydrophobic interaction chromatography (HIC) on an analytical column (TSK butyl-NPR 4.6 mm, 10 cm, and 2.5 mm, Tosoh Bioscience, South San Francisco, CA, USA). The HIC method used 1.5 M ammonium sulfate in 25 mM potassium phosphate pH 7.0 (mobile phase A) and 25 mM potassium phosphate pH 7.0 containing 25% isopropanol *v*/*v* (mobile phase B) run at a flow rate of 0.8 mL/min over a 12 min linear gradient with UV monitoring at 280 nm.

### 2.2. Radiochemistry

[^3^H]-MMAE was obtained from Seagen Inc. with a specific activity of 33.8 μCi/μg. [^3^H]-MMAE with the MC-vc-PAB linker with a specific activity of 19.9 μCi/μg (Seagen Inc., Bothell, WA, USA) was conjugated to mAb, as described in the above Section 2.1 [14]. The resulting [^3^H]-MMAE ADC had a specific activity of 33.3 μCi/mg and a DAR of 3.4 (Appendix A). The structures of the linker, drug, and the radiolabeled ^3^H location are shown in Figure 1.

The resulting [^3^H]-MMAE-polatuzumab vedotin was characterized for purity, DAR, binding, specific activity, etc., to ensure that the radiolabeling did not change these properties compared to the unlabeled materials before we dosed the animals. Polatuzumab vedotin and its unconjugated mAb were both radiolabeled with [^125^I] (non-residualizing) or [^111^In] with 1,4,7,10-tetraazacyclododecane-N,N′,N′′,N′′′-tetraacetic acid (DOTA) (residualizing) as described previously [15,16,17]. Briefly, the conjugation of [^125^I] or [^111^In]-DOTA was radiolabeled with intact polatuzumab vedotin (antibody with linker and payload).

For [^125^I] radiolabeling, sodium-^125^iodine was added to an iodination tube (Pierce, Rockford, IL, USA Cat# 28601) and, subsequently, conjugated to the polatuzumab vedotin. Then, to remove the excess unconjugated [^125^I], the solution was passed through a NAP^TM^-5, sephadex G-25 DNA grade, column (GE Healthcare Lifescience, Marlborough, MA, USA Cat# 17-0853-01). Similarly, DOTA was first conjugated to polatuzumab vedotin and purified with a NAP^TM^-5 column. The resulting DOTA-polatuzumab was then added to [^111^In] and chelated with ethylenediaminetetraacetic acid (EDTA) and purified with a NAP^TM^-5 column. The radiolabeled polatuzmab vedotin was characterized for purity, binding, specific activity, etc., before dosing the animals.

### 2.3. Animal Housing and Procedure

All animal studies were performed in the Genentech animal facility, which was accredited by the Association for Assessment and Accreditation of Laboratory Animal Care International. All the protocols and procedures for animal studies were approved (approval number: 12-2085, 12-2086, 14-0596, 14-2851, and 14-2852) by the Genentech Institutional Animal Care and Use Committee (IACUC) and performed in accordance with the institutional and regulatory guidelines. Rats in mass balance studies were kept in special metabolic cages (Lab Products Inc., Maywood, NJ, USA). Jugular-vein-cannulated, femoral-vein-cannulated, and bile-duct-cannulated rats were obtained from Charles River Laboratories (Hollister, CA, USA) after surgery and recovery. No special housing conditions were given for the animals in other studies. All the personnel involved in the animal experiments were trained according to IACUC guidelines.

### 2.4. Determination of the Tissue Distribution, Metabolism, and Elimination of Unconjugated MMAE in Rats

Twenty-one female (15 with jugular vein cannulation and 6 with bile duct cannulation (BDC)) 6–8-week-old Sprague Dawley rats, each weighing about 200 g, were obtained from Charles River Laboratories. Fifteen rats with jugular vein cannulation were given an intravenous (IV) bolus dose of [^3^H]-MMAE mixed with MMAE at 200 μg/kg (≈80 μCi/kg radioactivity level). Blood samples (≈0.3 mL) were collected from the jugular cannula at multiple time points post IV administration. To obtain plasma, blood was collected into lithium heparin tubes and gently inverted several times to properly mix the anticoagulant (lithium heparin). Tissue samples from different organs (e.g., the liver, heart, kidneys, lungs, etc.) were also collected from animals euthanized via exsanguination under anesthesia at the designated time points. Of the 15 rats, 6 rats with terminal time points at 1 and 7 days post-dose (*n* = 3 per terminal time point) were housed in metabolism cages for the collection of urine and feces.

The rats were housed in metabolism cages for one or two nights for acclimatization before dosing and were also kept in the metabolism cages post-dose. Urine and feces were collected from the rats at pre-dose, every 4 h from 0 to 8 h post-dose, once between 8 and 24 h, and then every 10 or 14 h until day 7 for the day 7 terminal time point animals. After each collection of urine and feces, the cages were rinsed with 10 mL of 50/50 methanol/water to remove any trace contaminants from previous collections. Another six rats with both a jugular vein cannula and a bile duct cannula were also given an IV bolus dose of [^3^H]-MMAE mixed with MMAE at 200 μg/kg (16 μCi/rat radioactivity level). Bile was collected through the bile duct cannula from animals for 7 days at designated time intervals for the purpose of metabolite profiling and identification.

Blood (plasma and cell pellet), tissues, urine, feces, and bile samples were processed using the method described previously in Shen et al. [6] for liquid scintillation counting. Briefly, tissues or feces samples were homogenized in a homogenization buffer (50 mM 4-(2-hydroxyethyl)-1-piperazineethanesulfonic acid (HEPES), 150 mM NaCl, and 10% glycerol) and a protease inhibitor cocktail (product number P8340, Sigma-Aldrich Corp, Saint Louis, MO, USA) using a probe-type tissue homogenizer (Tissumizer, S25N-8G metal probe, Teledyne Tekmar, Cincinnati, OH, USA). Each homogenate sample was then solubilized by mixing with SOLVABLE™ (No. 6NE9100, PerkinElmer, Shelton, CT, USA). The sample coloration was quenched using 30% H_2_O_2_ and the quenched samples were mixed with an Ultima Gold™ MV Liquid Scintillation Cocktail (No. 6013159, PerkinElmer, Walham, MA, USA). The total radioactivity of the processed samples was measured using a Tri Carb^TM^ 2900TR liquid scintillation analyzer (PerkinElmer, Walham, MA, USA).

### 2.5. Metabolite Profiling and Identification with Unconjugated MMAE in Rats

To characterize the metabolic profile of [^3^H]-MMAE, the bile samples were analyzed across time points that covered the majority of the radioactivity (up to 6 h with 75% of the total radioactivity injected) excreted. Due to the low radioactivity recovered in the urine, the urine samples were analyzed for up to 8 h. Both the bile and urine samples were investigated using Accela UPLC coupled with a Linear Trap Quadrupole (LTQ)-Orbitrap Velos system (Thermo Scientific, San Jose, CA, USA) and an online β-RAM 5C radiodetector (Lab Logics, Tampa, FL, USA) for radioprofiling.

Chromatographic separation was performed on a Luna C18 column (150 × 4.6 mm, 3 μm particle size, Phenomenex, Torrance, CA, USA) with mobile phases A (0.1% formic acid in water) and B (0.1% formic acid in acetonitrile) at a constant flow rate of 1 mL/min. The gradient was as follows: initial holding at 5% B for 2 min, increased to 15% B at 4 min, 42% B at 44 min, 75% B at 49 min, 95% B at 50 min, holding at 95% B until 55 min, decreasing to 5% B at 55.1 min, and then column re-equilibration until 60 min. The flow was split 3:1 post-column for radiomeasurements and mass spectrometry, respectively. As the majority of the metabolite was the intact [^3^H]-MMAE, only the intact [^3^H]-MMAE was quantified.

### 2.6. Determination of the Tissue Distribution, Catabolism, and Elimination of Polatuzumab Vedotin in Rats

The distribution of polatuzumab vedotin was investigated in the rats using the [^125^I]- and [^111^In]-conjugated to the antibody of polatuzumab vedotin or polatuzumab vedotin conjugated with [^3^H]-MMAE. In the ^125^I and ^111^In study, sixty femoral-vein-cannulated female Sprague Dawley rats aged 6–8 weeks (≈200 g) were obtained from Charles River Laboratories and were divided evenly into four groups (*n* = 15 each group). In the first group, a mixture of [^125^I]- and [^111^In]- polatuzumab vedotin was given as a tracer (5 μCi/rat, ≈0.05 mg/kg) IV bolus dose.

The second group of rats received an IV bolus dose of the radiolabeled tracer polatuzumab vedotin, along with a 10 mg/kg dose of unlabeled polatuzumab vedotin. For the third and fourth groups, the rats were dosed with [^125^I]- and [^111^In]- unconjugated polatuzumab antibodies instead of polatuzumab vedotin with the tracer and tracer plus 10 mg/kg unlabeled material, respectively. Blood and various tissues were collected at multiple intervals up to 14 days post-dose. To obtain the plasma, the blood was deposed into lithium heparin tubes and gently inverted several times to properly mix the anticoagulant. The total radioactivity of all samples was measured using a Perkin Elmer Wizard^2^ analyzer (PerkinElmer, Billerica, MA, USA).

The tissue distribution of the polatuzumab vedotin was also assessed using polatuzumab conjugated with [^3^H]-MMAE. Twenty-one female 6–8-week-old Sprague Dawley rats (15 normal rats and 6 rats with jugular vein and bile duct cannulation) weighing about 200 g each were obtained from Charles River Laboratories. Fifteen normal rats were given an IV bolus dose of polatuzumab vedotin conjugated with [^3^H]-MMAE at 10 mg/kg (30 μCi/rat radioactivity level). Blood samples (≈0.3 mL) were collected from the tail vein of the animals at the terminal time points post IV administration.

To obtain the plasma, the blood was collected into lithium heparin tubes and gently inverted several times to properly mix the anticoagulant. Tissue samples from different organs (e.g., the liver, heart, kidneys, and lungs) were also collected from animals euthanized via exsanguination under anesthesia at the designated time points. Of the 15 normal rats, 6 rats with terminal time points at 7 and 14 days were housed in metabolism cages for the collection of urine and feces to understand the mass balance and catabolite profiling. The rats were housed in metabolism cages for one or two nights for acclimatization before dosing and were also kept in the metabolism cages post-dose.

Urine and feces were collected from the rats at pre-dose, every 4 h from 0 to 8 h post-dose, once between 8 and 24 h, and then every 10 to 14 h until day 7 or day 14, depending on the animal’s terminal time point. After each collection of urine and feces, the cages were rinsed with 10 mL of 50/50 methanol/water to remove any trace contaminants from previous collections. Another six rats with both a jugular vein cannula and a bile duct cannula were also given an IV bolus dose of polatuzumab vedotin conjugated with [^3^H]-MMAE at 10 mg/kg (30 μCi/rat radioactivity level). Bile was collected through the bile duct cannula from animals for 14 days at designated time intervals.

The total radioactivity levels of the processed blood (plasma and cell pellet), tissues, urine, feces, and bile samples were measured using a Tri Carb^TM^ 2900TR liquid scintillation analyzer. The processed samples underwent protein precipitation with a 4:1 (*v*/*v*) acetonitrile-to-sample ratio to obtain the soluble radioactivity (presumed as deconjugated [^3^H]-MMAE or its metabolite(s)) and the precipitable radioactivity (presumed as [^3^H]-MMAE still conjugated to polatuzumab) measured using a Tri Carb^TM^ 2900TR liquid scintillation analyzer.

### 2.7. Catabolite Profiling and Identification with [^3^H]-Polatuzumab Vedotin in Rats

To characterize the catabolic profile of polatuzumab vedotin conjugated with [^3^H]-MMAE, bile and urine samples from the study with [^3^H]-MMAE polatuzumab vedotin were pooled across time points that covered >90% of the radioactivity excreted in that route and were investigated using a nanoACQUITY LC system (Waters, MA, USA) coupled with an LTQ-Orbitrap Velos system (Thermo Scientific, San Jose, CA) and an online β-RAM 5C radiodetector (Lab Logics, Brandon, FL, USA) for radioprofiling.

Chromatographic separation was performed on a Kinetex EVO C18 column (100 × 2 mm, 1.7 µm particle size, Phenomenex, Torrance, CA, USA) with mobile phases A (0.1% formic acid in water with 10 mM ammonium acetate) and B (0.1% formic acid in 90% acetonitrile in water with 10 mM ammonium acetate) at a constant flow rate of 0.1 mL/min. The gradient was as follows: initial holding at 10% B for 2 min, increased to 22% B at 4 min, 58% B at 45 min, 95% B at 50 min, holding at 95% B until 54 min, decreasing to 10% B at 54.5 min, and then column re-equilibration until 60 min. The flow was split 4:1 post-column for the radiomeasurements and mass spectrometry, respectively.

### 2.8. Data Analyses

To determine the percentages of the injected dose per gram or milliliter (%ID/g or %ID/mL) of the tissue/plasma/blood/excreta matrix in each sample, radioactivity data (counts per minute (cpm)/g or cpm/mL) were divided by the total radioactivity amount dosed to obtain the final values (in %ID/g or %ID/mL). To determine the total amount of radioactivity in each tissue, the total weight or volume of the tissue samples collected was used.

In instances where the total weight or volume could not be collected (tissues, such as muscle or blood), literature values were used to approximate the total sample weight or volume based on body weight. To determine the total amount of radioactivity or microgram equivalents (μg-Eq), radioactivity data (in cpm/g tissue or cpm/mL) were divided by the specific activity (in cpm/µg) of the dosing solution. All experiments were performed on biological replicates. The sample size for each experimental group is reported in the figure legends. The graphs were plotted and the statistical analyses were performed using Microsoft Excel version 16.34 (Microsoft, Redmond, WA, USA) and GraphPad Prism version 7.0 (GraphPad Inc., San Diego, CA, USA).

## 3. Results

### 3.1. Unconjugated [^3^H]-MMAE Showed Rapid Clearance from Systemic Circulation and Fast Tissue Distribution in Rats

The PK of unconjugated [^3^H] MMAE was first assessed following a single IV administration of 0.2 mg/kg [^3^H]-MMAE to rats. A rapid reduction of the MMAE radioactivity level was observed in the systemic circulation (Figure 2A). The whole-blood radioactivity was only 0.272 ± 0.0667 %ID/mL (0.109 ± 0.0267 μg-Eq/mL) at 2 min post-dose, which further decreased to 0.0668 ± 0.00605 %ID/mL (0.0267 ± 0.00242 μg-Eq/mL) at 10 min post-dose. A six- to eight-fold higher level of radioactivity was observed throughout the study in the cell pellets isolated from blood compared to plasma, suggesting a strong partition to red blood cells (Figure 2A).

As the MMAE radioactivity decreased from the systemic circulation, it was distributed to various tissues. The MMAE was quickly distributed to multiple organs, such as the liver, lungs, and kidneys within 10 min post IV administration. There is a large tissue partitioning in these highly perfused tissues as radioactivity detected in these tissues was greater than 1 %ID/g, a more than 15-fold higher level than detected in the whole blood. In contrast to the highly perfused tissues, there was little radioactivity detected in the brain (0.00738 ± 0.000937 %ID/g at 10-min post-dose). The radioactivity level peaked at 10 min in most tissues; however, the exposure of unconjugated MMAE in tissues could only be maintained for a short period as the radioactivity was barely detectable by 144 h (Figure 2B).

### 3.2. Unconjugated MMAE was Mainly Eliminated through Biliary/Fecal Routes with the Majority as Intact MMAE Together With Six Minor Metabolites Identified in Rat Bile Samples

To understand the elimination mechanism of unconjugated MMAE, a mass balance study was performed in rats. At 144 h post single IV administration, 98.3% (±6.00%) of the injected dose was excreted in the feces, while 8.61% (±3.16%) of the injected dose was recovered in the urine (Figure 3). A complete mass balance was achieved as approximately 100% was recovered in the feces and urine at 144 h post-dose. Intact MMAE was the only detectable species in the urine with no other metabolite detected.

In a separate study using bile-duct-cannulated rats to identify the metabolite, the collected bile samples showed similar radioactivity levels after 144 h, where 103% (±8.07%) of the injected dose was recovered. In the bile samples, six metabolites were identified (Figure 4), which included biotransformations, such as oxidation, demethylation, and amide hydrolysis. Intact MMAE remained as the major eliminated species (62.9% of the 74.2% total dosed radioactivity recovered in the bile up to 6 h post-dose).

### 3.3. Systemic Exposure of Radiolabeled Polatuzumab Vedotin Showed a Biphasic Elimination Profile Similar to That of Radiolabeled Unconjugated Polatuzumab Antibodies in Rats 

The systemic exposure of radiolabeled polatuzumab vedotin was then evaluated via labeling the antibody component with ^125^I and ^111^In and the MMAE component with ^3^H in Sprague Dawley rats, which is a non-binding species. First, the systemic exposure of [^125^I] and [^111^In]-polatuzumab vedotin (tracking mAb component) was compared to the [^125^I] and [^111^In]-unconjugated polatuzumab antibody. The [^125^I]-polatuzumab vedotin showed a similar biphasic elimination profile to that of the radiolabeled unconjugated polatuzumab antibody following a single IV dose at both dose levels of a tracer only (≈3 μg/kg) and a tracer plus 10 mg/kg of unlabeled materials (polatuzumab vedotin or unconjugated polatuzumab antibody, respectively) (Figure 5A). [^111^In]-polatuzumab vedotin also showed a similar biphasic elimination profile to that of [^125^I]-polatuzumab vedotin.

In the second study, in which a radiolabeled ^3^H-MMAE payload was conjugated to polatuzumab, the systemic exposure of [^3^H]-MMAE-polatuzumab vedotin also showed a biphasic elimination profile similar to the one with radioprobes tracking the antibody component (Figure 5B). In both studies, a biphasic systemic profile was observed, where the distribution phase lasted about 1 day, followed by a long elimination phase, which is typical of a humanized-IgG1 antibody distribution and elimination. In contrast to the direct dosing of unconjugated [^3^H]-MMAE, where the MMAE was partitioned with red blood cells (RBC), [^3^H]-MMAE-polatuzumab vedotin mainly existed in the plasma fraction of the blood instead of the RBC pellet (Figure 5B). Upon adding acetonitrile to the plasma samples (protein precipitation), over 95% of the radioactivity of [^3^H]-MMAE-polatuzumab vedotin was observed in the precipitated fraction, suggesting that the majority of MMAE remained conjugated to the antibody (Figure 5C).

### 3.4. Polatuzumab Vedotin Nonspecifically Distributed to Multiple Highly Perfused Tissues without Persistency Similar to Its Unconjugated Polatuzumab Antibody in Rats

The tissue distribution of polatuzumab vedotin was also investigated following a single IV injection of radiolabeled polatuzumab vedotin with ^125^I and ^111^In conjugated to the antibody of polatuzumab vedotin or polatuzumab vedotin with the [^3^H]-MMAE payload into rats (nonbinding species). While both ^125^I and ^111^In probes can be used to assess the tissue distribution, only ^125^I can be released back to the extracellular space after the intracellular degradation of the antibody, while ^111^In residualized inside cells as the DOTA cannot cross the cell membrane.

In the first study after [^125^I]- and [^111^In]-polatuzumab vedotin was administered to rats, multiple tissues were recipients of the distribution as rats are not a binding species, with relatively higher radioactivity distribution observed in well-perfused organs, including the lungs, heart, and kidneys (Figure 6A). This tissue distribution profile was similar to other hIgG1 monoclonal antibodies in nonbinding species, as described in previous manuscripts [4,18]. There was no specific tissue distribution, as all tissue radioactivity levels by tissue weight (%ID/g) were less than the whole blood by volume (%ID/mL). Co-dosing 10 mg/kg unlabeled polatuzumab vedotin, along with the radiolabeled polatuzumab vedotin, showed a similar radioactivity level and distribution profile (Appendix A), also suggesting that there was no specific distribution to a particular tissue.

When the radiolabeled polatuzumab vedotin was compared to unconjugated polatuzumab antibodies, a similar radioactivity level and tissue distribution profile were also obtained (Figure 6B) suggesting that there was a minimal impact from the conjugation of the payload. A 3-day post-dose skin sample in the unconjugated polatuzumab group appeared to have a particularly high level of radioactivity on both radioisotopes (as they were the same sample); thus, the average was higher for the skin at 3 days post-dose with a large standard deviation. While we suspected that this could be due to contamination (possibly blood contamination) or the sample size (as we did not collect all the skin but only a small piece), we did not further investigate the cause, as this sample appeared to be an outlier.

Finally, the ADC in tissues reached the peak radioactivity level at 1 or 24 h post administration and was then slowly eliminated, suggesting no persistency of polatuzumab vedotin in the tissues. The radioactivity level and tissue distribution profile were also similar when we tracked the ^3^H-MMAE payload conjugated to polatuzumab vedotin as compared to [^125^I]- and [^111^In]-polatuzumab vedotin (Figure 6C), indicating that the antibody component of the ADC, and not the payload of the ADC, dictated the distribution of the whole ADC.

To further understand the extent of the molecule internalization and catabolism, we have analyzed the difference between the ^111^In and ^125^I radioprobes. A side-by-side comparison of the radioactivity from these two probes in different tissues not only enabled us to monitor the tissue distribution but also helped to evaluate the organs where internalization and catabolism occurred (Figure 6A,B). In our results, the highest catabolism occurred in highly diffused tissues, such as liver, spleen, and kidneys. The results were mainly consistent between the ADC and the mAb at both tested concentrations (the ovaries and adrenal glands appeared to have higher catabolism in ADC but with a high variation).

To evaluate any possible release of [^3^H]-MMAE from the ADC, the total radioactivity of ^3^H in the soluble and precipitate fractions of the tissue homogenate was assessed after the addition of acetonitrile. While the majority of the [^3^H]-MMAE was observed in the precipitate fraction, the ^3^H radioactivity in the soluble fraction was detected in all different types of organs with very little in the plasma (Figure 6C). The soluble fraction represented the catabolites of polatuzmab vedotin as it released the [^3^H]-MMAE payload or ^3^H-containing catabolites slowly over time, and we could still observe a noticeable radioactivity level in the soluble fraction at 7 days post-dose.

### 3.5. Elimination and Catabolism of Polatuzumab Vedotin in Rats

To determine the route of elimination, we collected bile from BDC rats and feces and urine from normal rats over 14 days following the dosing with [^3^H]-MMAE-polatuzumab vedotin, and we analyzed the total radioactivity in the excreta. Biliary/fecal secretion was identified as the major route of elimination for [^3^H]-MMAE-polatuzumab vedotin, of which 103% (±11.8%) of the total injected radioactivity was recovered in the feces samples, while only 5.21% (±1.03%) was found in the urine (Figure 7). Greater than 80% of the ^3^H radioactivity recovered in the urine/feces was in the acetonitrile soluble fraction. The mass balance was achieved as approximately 100% of the injected dose was recovered in the feces and urine.

In the separated BDC rat study, the collected bile samples accounted for about 67% of the injected radioactivity dose, which corroborated with the feces data in the normal rat study. The lower recovery of radioactivity in BDC rats was likely due to the deteriorating health of the BDC rats after surgery. In these collected bile and urine samples, a total of eight catabolites (C1 to C8) of the polatuzumab vedotin were observed through liquid chromatography and mass spectrometry profiling with an online radiodetector, including MMAE (C7). The structures of six catabolites were identified and are summarized in Table 1 and shown in Figure 8. Five catabolites were among the metabolites found in the unconjugated [^3^H]-MMAE study. The most abundant catabolite detected was the unconjugated MMAE (C7), followed by C4, C5, and C3, respectively.

## 4. Discussion

ADCs are a therapeutic class that utilizes a highly potent cytotoxic payload that has a narrow therapeutic window if these cytotoxic payloads are administered alone. Upon the conjugation of cytotoxic payloads to a monoclonal antibody via a linker, the ADC improves in specificity, allowing for specific delivery to a tumor target and reducing the systemic exposure and normal tissue uptake, thus, increasing the therapeutic window. As such, it is important to gain ADME knowledge of the ADC and its cytotoxic payload.

This study used a radiolabeling technique in a systematic approach with a preclinical species, namely, rats, to answer key questions related to the distribution, catabolism/metabolism, and elimination of polatuzumab vedotin and its cytotoxic payload, namely, MMAE, as a case study to aid in clinical development. Since an ADC contains multiple components, we took both large molecule and small molecule features into consideration for ADME studies by radiolabeling each component, as previously described [6,13], where we tracked the MMAE component by incorporating ^3^H into the payload, while the antibody component, namely, polatuzumab, was radiolabeled with ^125^I or ^111^In.

Polatuzumab vedotin utilized a protease cleavable MC-vc-PAB linker joining a polatuzumab antibody and an MMAE cytotoxic payload. Upon internalization into the cell, we expected that MMAE would be released from the ADC via the cleavable linker via cathepsin B or through proteolytic degradation [19]. Therefore, we first characterized the DME properties of unconjugated MMAE in rats, which was the species used for the PK and toxicology studies. Following a single IV administration, unconjugated [^3^H]-MMAE exhibited fast systemic clearance and showed a wide distribution into multiple organs, as expected. Highly perfused organs, such as the liver, kidneys, and lungs, exhibited a higher distribution, while poorly perfused tissues, such as the muscles and skin, showed a lower amount of radioactivity. Interestingly, the brain showed minimal exposure, likely because it was protected by the blood–brain barrier. Rapid tissue distribution was followed by fast elimination with no persistency observed in any tissue examined in the study. Upon further analysis of the radioactivity in the plasma and cell pellet fraction, we found that the radioactivity level in the blood was six- to eight-fold higher than in the processed plasma, indicating that MMAE had high red blood cell partitioning, which is consistent with the results from the in vitro assessment (manuscript under preparation).

While MMAE’s high blood partitioning can act as a depot for MMAE to remain systemically, the level of MMAE in the blood was much lower compared to the tissues we examined, and thus, its contribution to the tissues was minimal. The major elimination route of the IV administered unconjugated MMAE was through biliary–fecal secretion, where 98.3% (±6.00%) of the injected dose was excreted in the feces, whereas only 8.61% (±3.16%) of the injected dose was recovered in the urine over 7 days. Six metabolites were also observed and identified in the bile with unchanged MMAE as the primary species was eliminated, which accounted for 62.9% of the 74.2% total dosed radioactivity recovered in the bile up to 6 h post-dose.

The distribution, catabolism, and elimination of polatuzumab vedotin was investigated in the next set of studies. The method we employed to track the antibody component of the ADC was through ^125^I or ^111^In radiolabeling of the antibody. Following an IV administration of [^125^I]- or [^111^In]-polatuzumab vedotin or an unconjugated polatuzumab antibody, the plasma radioactivity–time profile followed a biphasic elimination with a short distribution phase, followed by a long elimination phase that was similar to the PK studies described by Li et al. [4]. When the polatuzumab vedotin plasma radioactivity–time profile was compared to that of unconjugated [^125^I]- or [^111^In]-polatuzumab antibodies, both profiles showed a similar pattern.

However, the radioactivity level of the unconjugated polatuzumab antibody was higher than that of polatuzumab vedotin, which demonstrated that the systemic clearance of ADCs tended to be faster than that of the unconjugated mAbs, which was attributed to the “impact of conjugation” [15]. Similarly, the tissue distribution profile of polatuzumab vedotin and its unconjugated mAb were comparable, with higher radioactivity distribution observed in well-perfused organs, including the lung, heart, and kidney (radioactivity levels were slightly different between the two). Taken together, these data indicated that the conjugation of MMAE to polatuzumab did not severely alter the distribution profile as compared to the unconjugated polatuzumab antibody.

To track the fate of the cytotoxic payload, [^3^H]-MMAE was conjugated to polatuzumab mAb to form the [^3^H]-MMAE-polatuzumab vedotin. After an IV administration, [^3^H]-MMAE-polatuzumab vedotin followed a biphasic elimination pattern similar to what was seen for [^125^I]- or [^111^In]-polatuzumab vedotin (tracking mAb component) but very different from the profile of unconjugated [^3^H]-MMAE, where [^3^H]-MMAE was cleared systemically within hours. This data suggested that the conjugation of [^3^H]-MMAE to the antibody could retain the exposure of MMAE, allowing for better tissue targeting, such as tumors, for the uptake.

Further analysis of the plasma using organic solvent extraction revealed that over 99% of the ^3^H radioactivity in the plasma from the ADC dose was precipitable, indicating that the payload remained conjugated to the antibody with good linker stability. The tissue distribution profile of [^3^H]-MMAE-polatuzumab vedotin demonstrated a similar pattern to that of [^125^I]- or [^111^In]-polatuzumab vedotin and -unconjugated mAb, further confirming that the stability of ADC, as well as the antibody component, drove the ADC tissue disposition.

When the tissue homogenates were further analyzed using organic solvent extraction, there were different levels of soluble radioactivity in the tissues, which were, in general, higher than those seen in the plasma, suggesting that [^3^H]-MMAE or other ^3^H-containing catabolites were released in the tissues. The soluble radioactivity detected in the tissue dosed with [^3^H]-MMAE-polatuzumab vedotin decreased after the initial peak and throughout the course of the study without persistency over time, except in the dorsal root ganglion (the dorsal root ganglion was not collected in the unconjugated [^3^H]-MMAE study). The prolonged exposure of MMAE in the dorsal root ganglion might have contributed to peripheral neuropathy, which was reported as one of the toxic side effects of the MMAE–ADC treatment [20].

The elimination of [^3^H]-MMAE-polatuzumab vedotin was mainly through biliary fecal secretion, with 103 ± 11.8% of the injected dose recovered in the feces and 5.21 ± 1.03% in the urine. Organic solvent precipitation of the feces samples resulted in over 80% of the radioactivity residing in the soluble fraction, suggesting that a catabolite of [^3^H]-MMAE-polatuzumab vedotin, such as deconjugated [^3^H]-MMAE or its downstream metabolite, were eliminated in a similar fashion to the unconjugated [^3^H]-MMAE when directly dosed. Further evidence of this claim was shown when we characterized the catabolites found in urine and bile (fecal–biliary path).

There were eight catabolites detected in the urine and bile samples dosed with [^3^H]-MMAE-polatuzumab vedotin, of which seven were identified. C7, which was identified as unconjugated MMAE, was the most abundant catabolite detected, while five other catabolites (C2 through C6) were also identified as metabolites from the unconjugated [^3^H]-MMAE study. C8, which was identified as cys-vc-MMAE, was the only catabolite not found in the unconjugated [^3^H]-MMAE study. Since unconjugated MMAE was the main catabolite eliminated over a long period, the role of the CYP enzyme in the metabolism would be minimal and, thus, would reduce the risk of major DDI adverse effects.

As the liver played a critical role in the elimination of MMAE in rats, patients with hepatic impairment may experience an accumulation of MMAE in the body, leading to safety concerns. Limited mass balance data was reported by Han et al., who administered brentuximab vedotin to patients and found that MMAE was mostly eliminated as intact and through the feces [21], which corroborated with our result as brentuximab vedotin utilized the same linker and payload as polatuzumab vedotin. Renal elimination was only a minor component and, thus, there was a low risk of renal impairment and might not warrant additional study.

## 5. Conclusions

In summary, we systematically characterized the distribution, catabolism, and elimination of polatuzumab vedotin and its unconjugated MMAE payload in rats using multiple radiometric approaches. Our studies demonstrated that polatuzumab vedotin had a stable linker in circulation and confirmed that the antibody component of the ADC dictated the overall ADC disposition in rats. Upon the distribution/internalization into the tissues, ADC underwent catabolism to release MMAE and several other catabolites over time, which were mainly eliminated via the biliary–fecal route, with a minor fraction through the urine in rats.

These results imply that the liver plays a more important role than the renal clearance in the elimination of polatuzumab vedotin catabolites; therefore, additional hepatic impairment studies in patients might provide clinicians with further insight. Though unconjugated MMAE was identified as the major catabolite, the overall DDI risk was low given that the systemic and liver unconjugated MMAE levels were low, in conjunction with other assessments (in vitro DDI study not shown in this manuscript). In conclusion, these studies not only provide information to support the clinical development of polatuzumab vedotin but also provide a comprehensive and systematic approach to characterizing the ADC DME properties, which may help the development of other ADC therapeutics.

## Figures and Tables

**Figure 1 jcm-10-01323-f001:**
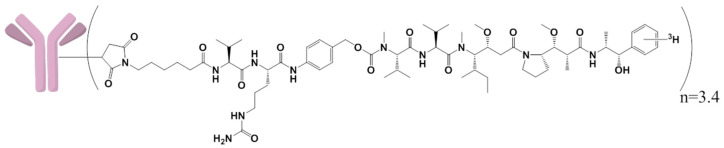
Structure of [^3^H]-monomethyl auristatin E (MMAE)-polatuzumab vedotin (drug-to-antibody ratio (DAR) of 3.4) and the location of the radiolabeled ^3^H on the MMAE.

**Figure 2 jcm-10-01323-f002:**
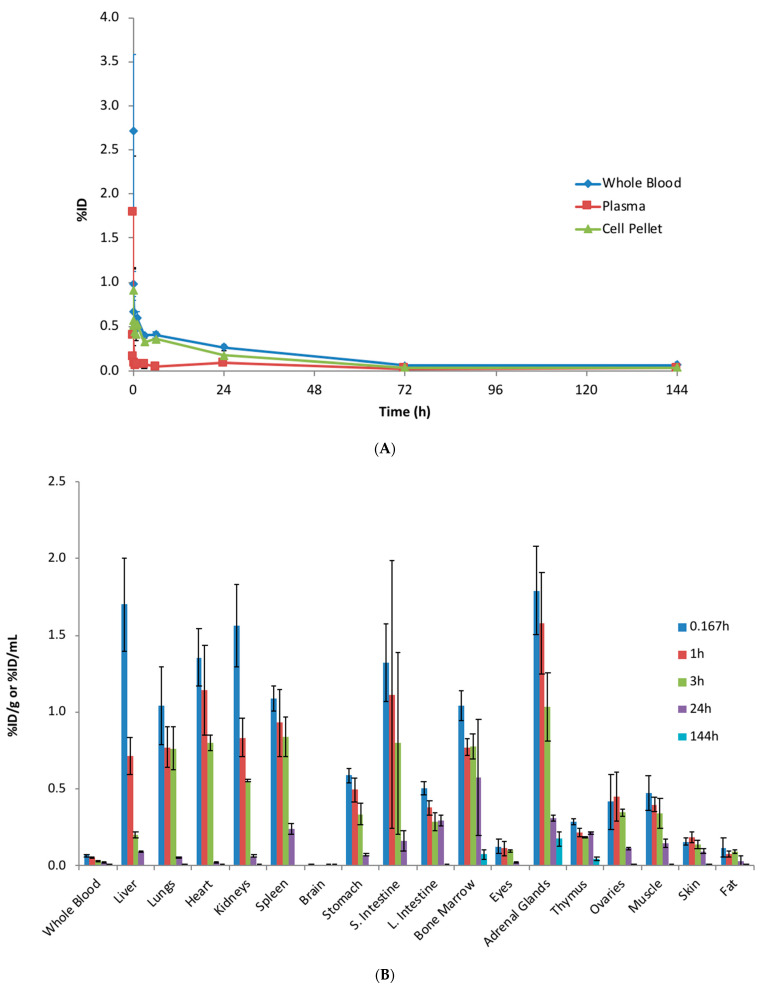
The systemic pharmacokinetics (PK) and tissue biodistribution of unconjugated [^3^H]-MMAE following a single intravenous (IV) administration. (**A**) Radioactivity of [^3^H]-MMAE (as a percentage of the injected dose) from the whole blood, plasma, and cell pellets in rats up to 144 h post administration. A six- to eight-fold higher level of radioactivity was observed throughout the study in the cell pellets isolated from blood compared to the plasma, suggesting a strong partition to red blood cells. (**B**) Radioactivity of [^3^H]-MMAE from blood and multiple tissues isolated from rats at various time points post IV administration. The data are represented as the mean ± standard deviation (SD). *n* = 3 per time point. S. Intestine: small intestine; L. Intestine: large intestine.

**Figure 3 jcm-10-01323-f003:**
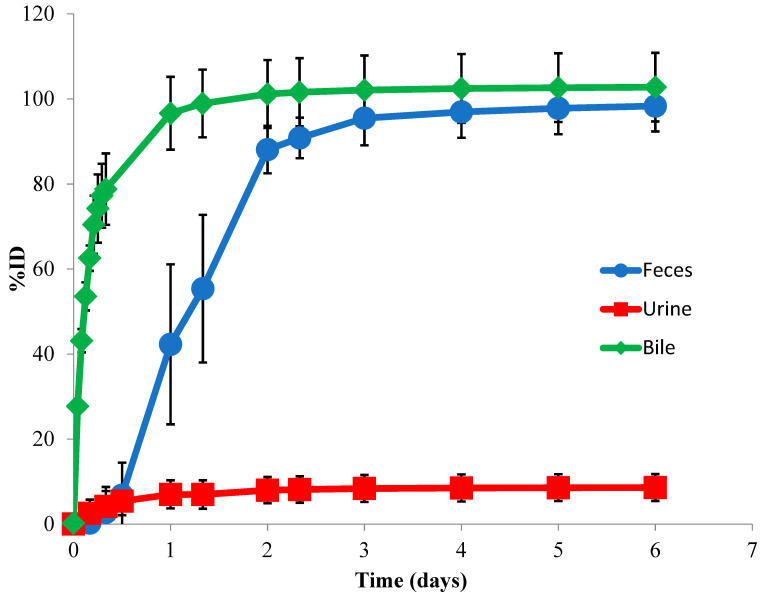
The route of elimination of unconjugated [^3^H]-MMAE in rats following a single IV administration. The total ^3^H radioactivity in feces and urine samples was collected through the study, while bile samples were collected in a separate set of bile-duct-cannulated rats. The data are represented as the mean ± SD. *n* = 6 per time point up to 24 h (including 24 h), and *n* = 3 per time point beyond 24 h to the end of the study.

**Figure 4 jcm-10-01323-f004:**
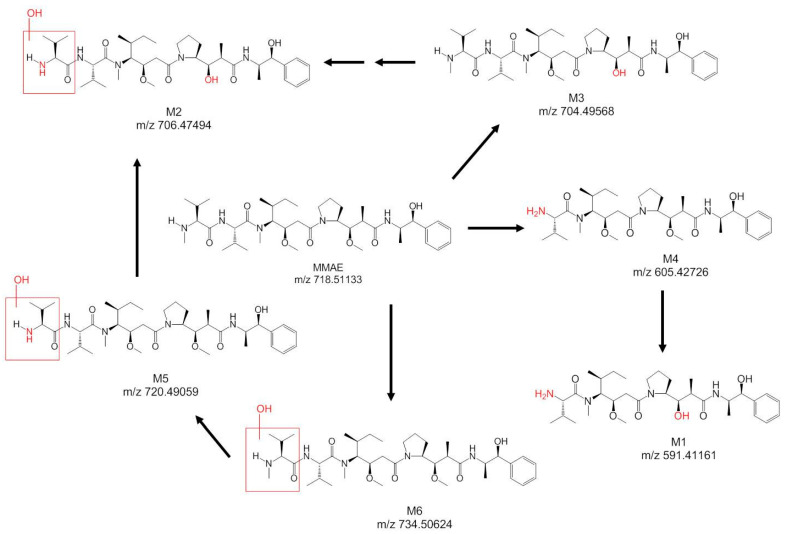
Structure of MMAE and its metabolites identified, with the possible biotransformations in the rat bile samples dosed with [^3^H]-MMAE.

**Figure 5 jcm-10-01323-f005:**
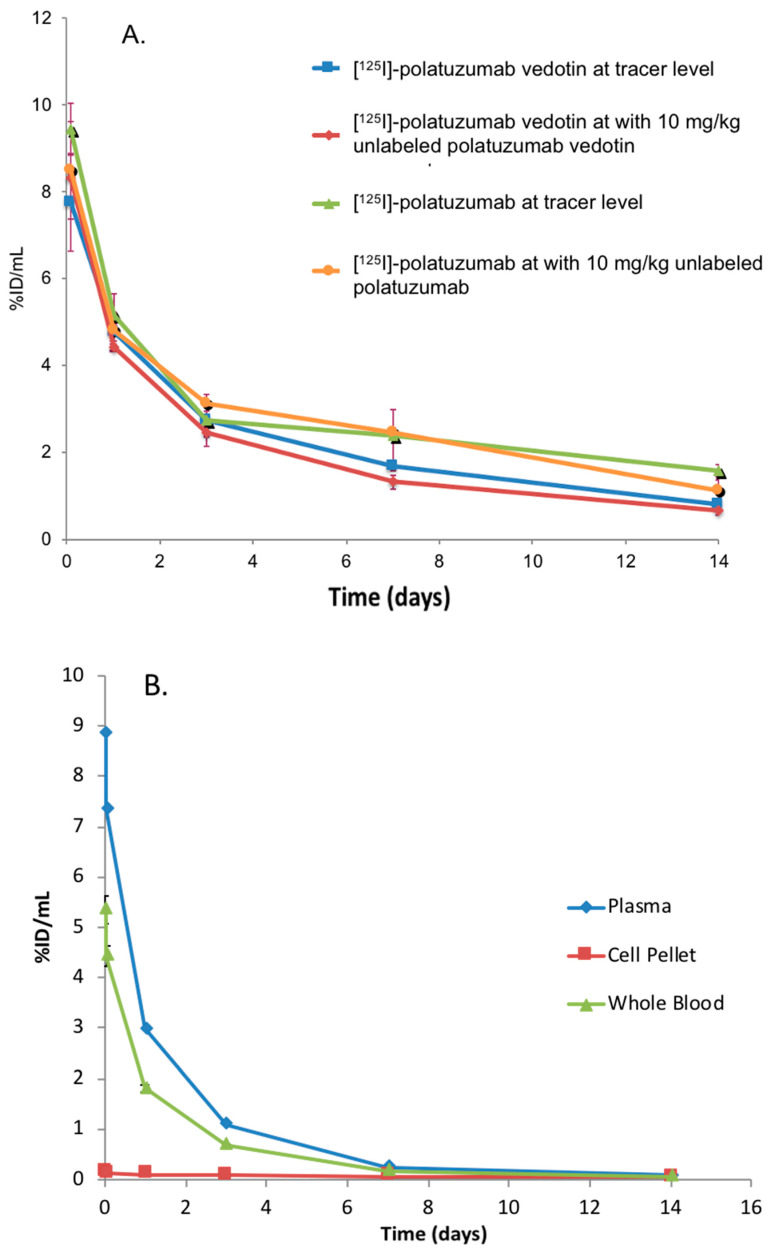
Profiles of the blood radioactivity levels vs. time of radiolabeled polatuzumab vedotin following a single IV administration. (**A**) Radioactivity of [^125^I]-polatuzumab vedotin (as a percentage of the injected dose) comparing with [^125^I]-polatuzumab antibodies in plasma in rats at tracer (≈3 ug/kg) and tracer plus 10 mg/kg dose levels up to 14 days post administration. The data are represented as the mean ± SD. *n* = 3 per time point. (**B**) Radioactivity of [^3^H]-MMAE-polatuzumab vedotin (as a percentage of the injected dose) from the whole blood, plasma, and cell pellets in rats up to 14 days post administration. The data are represented as the mean ± SD. *n* = 3 per time point. (**C**) Plasma from the rats dosed with [^3^H]-MMAE-polatuzumab vedotin were further processed using organic solvent precipitation, which showed that most of the radioactivity was in the precipitable fraction, suggesting that the MMAE payload was still conjugated to the antibodies in the plasma.

**Figure 6 jcm-10-01323-f006:**
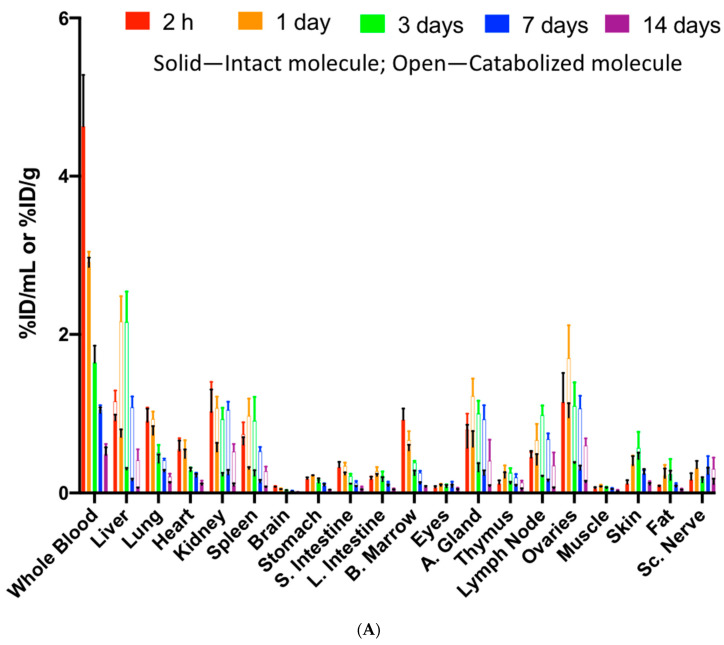
The tissue distribution of radiolabeled polatuzumab vedotin and unconjugated polatuzumab antibody after a single dose via IV administration in rats. (**A**) The tissue and blood radioactivity from intact (^125^I radioactivity—solid bar) and catabolized (subtract ^125^I radioactivity from ^111^In radioactivity—open bar) polatuzumab vedotin dosed at the tracer (≈3 μg/kg) level up to 14 days post administration. (**B**) The tissues and blood radioactivity from intact (^125^I radioactivity—solid bar) and catabolized (^111^In subtracted with ^125^I radioactivity—open bar) unconjugated polatuzumab antibody dosed at the tracer (≈3–4 μg/kg) level up to 14 days post administration. (**C**) The tissues and plasma radioactivity from conjugated (^3^H precipitated radioactivity—solid bar) and deconjugated (^3^H soluble radioactivity—open bar) [^3^H]-MMAE-polatuzumab vedotin dosed at 10 mg/kg up to 14 days post administration. A higher distribution of polatuzumab vedotin was observed in highly perfused tissues, such as the liver, spleen, and kidneys, and resulted in higher catabolism. The data are represented as the mean ± SD (except in (**C**) where only the mean is reported). *n* = 3 per time point. S. Intestine: small intestine; L. Intestine: large intestine; B. Marrows: bone marrows; A. gland: adrenal glands; DRG: dorsal root ganglions; Sc. Nerve: sciatic nerve.

**Figure 7 jcm-10-01323-f007:**
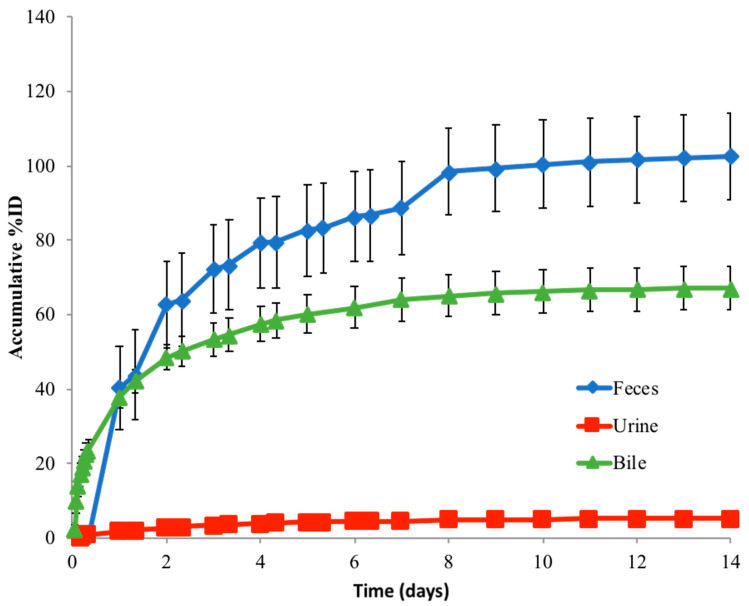
The route of elimination of [^3^H]-MMAE-polatuzumab vedotin in rats following a single IV administration. The total ^3^H radioactivity in the feces and urine samples collected through the study. Bile samples were collected in a separate set of bile-duct-cannulated rats. The acetonitrile (ACN) precipitation of urine and feces samples showed that the majority (>80%) of radioactivity was in the ACN-soluble fraction, suggesting that [^3^H]-MMAE-polatuzumab vedotin was eliminated mainly as [^3^H]-MMAE or [^3^H]-containing small molecule catabolites. Similarly, the acetonitrile precipitation of bile samples showed >95% of the radioactivity in the soluble fraction. The data are represented as the mean ± SD. For the urine and feces data, *n* = 6 per time point up to 7 days (including at day 7), *n* = 3 per time point from beyond 7 days to the end of the study. *n* = 6 per time point for the bile samples.

**Figure 8 jcm-10-01323-f008:**
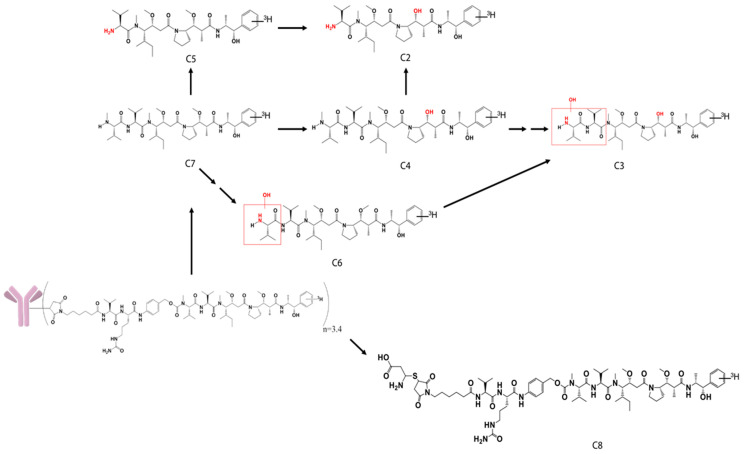
Structure of the catabolites identified, with the possible biotransformations in rat bile samples dosed with [^3^H]-MMAE-polatuzumab vedotin.

**Table 1 jcm-10-01323-t001:** Catabolites of [^3^H]-MMAE-polatuzumab vedotin identified in the Sprague Dawley rats.

Catabolite	Protonated Molecular Formula	Theoretical *m*/*z* for [M+H]^+^	Measured *m*/*z* for [M+H]^+,b^	Mass Accuracy (ppm)	%ID in Bile	%ID in Urine
C1	NA	NA	NA		NA	2.26
C2	C_32_H_55_N_4_O_6_^+^	591.4116	591.4110	−1.12	3.94	NA
C3	C_37_H_64_N_5_O_8_^+^	706.4749	706.4749	−0.13	9.32	NA
C4	C_38_H_66_N_5_O_7_^+^	704.4957	704.4958	0.16	13.49	NA
C5	C_33_H_57_N_4_O_6_^+^	605.4273	605.4264	−1.44	12.16 ^c^	NA
C6	C_38_H_66_N_5_O_8_^+^	720.4906	720.4905	−0.15		
C7	C_39_H_68_N_5_O_7_^+^	718.5113	718.5112	−0.22	16.48 ^d^	2.56 ^d^
C8	C_71_H_114_N_12_O_17_S^2+^	719.4067 ^a^	719.4047 ^a^	−2.88	2.60	NA

ID: injected dose; NA: not applicable. ^a^ C8 *m*/*z* corresponds to the doubly charged species. ^b^ The *m*/*z* values reported for MMAE and its catabolites were from the most abundant ion incorporating one ^3^H. ^c^ Catabolites C5 and C6 shared the same retention time. Therefore, the combined total %ID of catabolites C5 and C6 is listed in the table. ^d^ Catabolite C7 was detected in three matrices: plasma, bile, and urine. The detected amount of C7 in the plasma was too low to calculate a percentage of the ID.

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
