# Peer review of "Preclinical Characterization of the Distribution, Catabolism, and Elimination of a Polatuzumab Vedotin-Piiq (POLIVY®) Antibody–Drug Conjugate in Sprague Dawley Rats"

_jcm, 2021, doi:10.3390/jcm10061323_

Round 1
Reviewer 1 Report
This original manuscript submitted to the JCM by Yip et al., described the antibody-drug conjugate (ADC; Polatuzumab vedotin (or POLIVY®) ) composed of a polatuzumab mAb conjugated to monomethyl auristatin E (MMAE) through a protease cleavable dipeptide linker, that has been approved by the FDA for the treatment of diffuse large B-cell lymphoma (DLBCL). Authors characterize and compared the distribution, catabolism and elimination of ADC, its Mab, and MMAE payload in rats using three different radioisotopes. The experiment set-up is simple and straightforward approach in pre-clinical model for the evaluation of ADME. However, authors did not provide many information including linker peptide, lack of conjugation details on conjugation chemistry and in vitro characterization.
Comments:
- It is good this work describes the ADME, however, authors did not provide information of what percent of polatuzumab, and its ADC internalize into the cells? Authors should provide in vitro study results?
- Please provide more information on MC-vc-PAB linker and reference
- How average drug‐to‐antibody ratio (3.4-3.5) was assessed please provide data?
- How DOTA was conjugated to Polatuzumab vedotin, first DOTA conjugated and then MMAE or with ADC it is confusing
- I think M&M was pretty short please provide more information
- Authors should provide binding study data of ADC and its antibody after conjugation which is pretty routine data prior to perform animal studies
- Line #324; “The radiolabeled-polatuzumab vedotin showed a similar biphasic elimination profile to that of radiolabeled-unconjugated polatuzumab antibody..” authors referring which isotope? Please mention specifically, (125-I vs 111-In) or make it clear for example 125- Polatuzumab vedotin or 111-In-Polatuzumab vedotin.
- Please change the figure 5 legends with specific tracer name of the isotope labeled (instead of Polatuzumab vedotin tracer level) for with and without unmodified antibody.
- Figure 6B, why skin uptake is very high?
Author Response
Comment on English Language and Style:
Response: We have sent our manuscript to MDPI Journal Editor for English language and style improvement.
Comments:
1. It is good this work describes the ADME, however, authors did not provide information of what percent of polatuzumab, and its ADC internalize into the cells? Authors should provide in vitro study results?
Response:
We acknowledge the comment. The aim/focus of this study was to characterize the ADME properties of POLIVY in a non-binding species (rats), so to help assess the potential need for drug: drug interaction (DDI) and special patient population (such as with hepatic or renal impairment) studies. The information about the pharmacology including the internalization of polatuzumab and its ADC into the cells have been described in the previous publication by Polson et al. (2007, referenced in this paper). We have added this information to the introduction in page 5 of the manuscript.
2. Please provide more information on MC-vc-PAB linker and reference
Response:
Though the detail information about the “MC-vc-PAB linker” platform has been described in multiple publications, for the convenience of the readers, we have now added brief description on the “MC-vc-PAB linker” in the introduction on page 4 of the manuscript and also included a reference by Jain et. al (2015) who illustrated the details of MC-vc-PAB linker technology.
3. How average drug‐to‐antibody ratio (3.4-3.5) was assessed please provide data?
Response:
The Drug-to-antibody ratio (DAR) was calculated based on the analysis of hydrophobic interaction chromatography (HIC) HPLC. The details of the HIC-HPLC method was added to Section 2.1 in M&M on page 8. The analytical data of [3H]-MMAE-polatuzumab vedotin was added in the supplemental Figure 1.
4. How DOTA was conjugated to Polatuzumab vedotin, first DOTA conjugated and then MMAE or with ADC it is confusing
Response:
MMAE was first conjugated to the antibody to make polatuzumab vedotin (i.e., ADC), then DOTA was conjugated to the antibody component of the ADC, followed by adding [111In] to DOTA. Similarly, [125I] was conjugated to polatuzumab vedotin ADC after MMAE conjugation. We have added clarification to Section 2.2 on page 8-9.
5. I think M&M was pretty short please provide more information
Response:
We originally intended to keep the manuscript concise by referencing several published manuscripts for the M&M such as ADC conjugation, sample processing and HPLC methods as the same M&M were used in these references. Per Reviewer’s suggestion, we have now expanded the M&M sections. (Page 8-11, 14).
6. Authors should provide binding study data of ADC and its antibody after conjugation which is pretty routine data prior to perform animal studies
Response:
We acknowledge the comment. It is a bit unclear whether the Reviewer is asking about the impact of MMAE conjugation on antibody binding or the impact of radiolabeling/conjugation on binding. If it is for the first case, Polson et al. have shown that ADC and its unconjugated antibody had very comparable binding to its target and internalization (data in Polson et al., 2007). We have added some information from the Polson’s paper in our manuscript (page 5). If the Reviewer is asking whether the radiolabeling would impact the binding, we have routinely characterized the quality of the radiolabeled compounds including purity, DAR, binding, and specific activity to ensure the radiolabeling did not change these properties as comparing to the unlabeled materials before we dose them into animals (we have not shown all these characterization data as this manuscript is more about ADME investigation). In addition, for the current study, we have used a non-binding specifies (rats) for the ADME studies. The tissue uptake is mainly driven by the non-specific mechanism where the target binding is not a concern.
7. Line #324; “The radiolabeled-polatuzumab vedotin showed a similar biphasic elimination profile to that of radiolabeled-unconjugated polatuzumab antibody..” authors referring which isotope? Please mention specifically, (125-I vs 111-In) or make it clear for example 125- Polatuzumab vedotin or 111-In-Polatuzumab vedotin.
Response:
Both [125I] and [111In] radiolabeled polatuzumab vedotin has shown very similar biphasic elimination profile. In order to make the manuscript concise, we only showed the [125I] data in the manuscript. We have now modified the text to clarify this point (Page 21-22).
8. Please change the figure 5 legends with specific tracer name of the isotope labeled (instead of Polatuzumab vedotin tracer level) for with and without unmodified antibody.
Response:
We have made the changes to show the radio-isotope used so to clarify the graph (Page 23).
9. Figure 6B, why skin uptake is very high?
Response:
For that particular time point (day-3 post-dose), one of the three skin samples had a particular high reading on both radio-isotopes (both came from the same sample), thus, the average was higher with a large standard deviation. Nevertheless, the overall radioactivity uptake in the skin remained lower than that in highly perfused tissues such as the liver and kidneys. While we suspect this could be due to a cross-contamination (possibly blood contamination) or sample size (as we did not collect all the skin but only a small piece), we did not further investigate the cause as this sample seems like an outlier and does not impact data interpretation. We have modified the text to clarify this point (Page 25-26).
Reviewer 2 Report
The readers of this journal will be very intrested in this paper.
Author Response
Comment on English Language and Style:
Response: We have sent our manuscript to MDPI Journal Editor for English language and style improvement.
Reviewer 3 Report
The authors addressed the important issue of ADC elimination pathways. ADC represents novel class of anticancer drugs. There are many unknowns about how such complicated molecules and their catabolites distribute and accumulates in the tissues. In this manuscript authors using radiolabeled molecules systemically characterized distribution, metabolism/catabolism, and elimination of polatuzumab vedotin (POLIVY), an antibody-drug conjugate (ADC). Interestingly polatuzumab vedotin showed a biphasic elimination profile. Moreover presented studies shows new directions for potential drug-drug interaction.
Minor remark:
Figure 2A
Markers on Y axis should be corrected and decription below X axis spread/corrected.
Author Response
Comment on English Language and Style:
Response: As mentioned above, we have sent our manuscript to MDPI Journal Editor for English language and style improvement.
Comment:
- Figure 2A - Markers on Y axis should be corrected and description below X axis spread/corrected.
We have updated Figure 2A on page 17. The standard deviation was the same color as the Y-axis marker, as such, it looked like it was mis-aligned. X-axis spread has also been corrected.
Round 2
Reviewer 1 Report
Authors addressed all the questions and comments adequately.
Hence, I would like to recommend accept for the publication.
